# The Development of a Procedure for the Cryopreservation of the Callus of *Anthurium andraeanum* by Vitrification

**DOI:** 10.3390/plants13213106

**Published:** 2024-11-04

**Authors:** Yiying Zhang, Shan Deng, Huifeng Lin, Yunxia Chu, Jingyan Huang, Shouguo Li, Fazhuang Lin, Sumei Zhang, Weilan Jiang, Li Ren, Hairong Chen

**Affiliations:** 1Institute for Agri-Food Standards and Testing Technology, Shanghai Academy of Agricultural Sciences, No. 888, Rd. Yezhuang, Shanghai 201403, China; zyy425zoey@163.com (Y.Z.); dengshan85@163.com (S.D.); chuyx@189.cn (Y.C.); yanyandcxc@163.com (J.H.); lsgwjh@outlook.com (S.L.); 2Flower Research Institute, Sanming Academy of Agricultural Scienses, No. 2, Str. Qiujiang Zhuyuan, Shaxian, Sanming 365051, China; linhueifeng@163.com (H.L.); lfz0lfz@163.com (F.L.); 3College of Bioscience and Biotechnology, Yangzhou University, No. 48, Rd. Wenhui, Yangzhou 225009, China; 17385980450@163.com; 4School of Agriculture, Anshun University, No. 25, Rd. Xueyuan, Anshun 561000, China; jwc4024@163.com

**Keywords:** Anthurium andraeanum, callus culture, cryopreservation, vitrification

## Abstract

The cryopreservation of *Anthurium andraeanum* germplasm resources is extremely important for the production and selection of new varieties. At present, the cryopreservation procedure for the callus of *A. andraeanum* has not been established. In this study, the leaves of *A. andraeanum* were used as explants to culture the callus. The cryopreservation procedure of the callus by vitrification was initially established by using the orthogonal experimental method of four factors and three levels in the preculture, loading, and dehydration steps. Furthermore, the vitrification-based cryopreservation was optimized by changing the preculture temperature and loading solution and adding exogenous substances to the plant vitrification solution (PVS2). In this procedure, the callus was precultured at 25 °C for 2 d, and loaded in 50% PVS2 at 25 °C for 60 min. The callus was dehydrated with PVS2 containing 0.08 mM reduced glutathione (GSH) at 0 °C for 60 min. After rapid-cooling in liquid nitrogen for 1 h, it was rapid-warming in a water bath at 40 °C for 90 s and unloaded for 30 min. After 1 d of recovery, the cell relative survival rate of the cryopreserved callus was 64.60%. The results provide a valuable basic and effective method for the long-term conservation of *A. andraeanum* germplasm resources.

## 1. Introduction

*Anthurium andraeanum* Lind. is a perennial evergreen herbaceous plant in the genus *Anthurium* of the family Araceae, originating from Costa Rica, Colombia, and other tropical rainforest areas [1], and has high economic and ornamental value because of its bright and colorful spathe. Nowadays, with the continuous development of plant introduction technology, *A. andraeanum* is widely cultivated in Europe, Asia, Africa, and other regions. *A. andraeanum* varieties are faced with intolerance to low temperatures and the high costs of environmental control using greenhouse production due to their origin in tropical rainforest regions [2]. The work of traditional germplasm preservation requires a lot of human, material, and financial resources. The in vitro culture technique is a biotechnology in which the tissues, organs, and cells of plants are cultured in vitro to obtain a large number of regenerated plants, which has the characteristics of a wide range, high efficiency, and good quality. Young leaves had the highest induction rate and the lowest contamination rate in the in vitro culture of *A. andraeanum* [3]. However, in vitro tissue culture requires non-stop succession culture, and its genetic traits are prone to mutation. Therefore, it is of great significance to research a new method to preserve the germplasm resources of *A. andraeanum* for a long time in the production and the selection and breeding of new varieties.

Cryopreservation is the most promising method for the long-term effective and stable preservation of plant cells, tissues, or organs at an ultra-low temperature of −196 °C, generally using liquid nitrogen (LN) [4,5]. Cryopreservation can stop intracellular metabolism, not only to ensure that the cells, after freezing, maintain normal biological activity and genetic stability but also to avoid genotypes and chromosomal mutations in long-term succession processes. Sakai [6] first reported the use of cryopreservation for freezing and storing mulberry materials. After decades of research and development, different types of plant materials, including reproductive organs [7,8], nutrient organs [9], tissue cultures [10], etc., have been used for cryopreservation. Currently, more than 10,000 resources have been preserved in cryopreservation repositories established by various countries around the world [11,12].

Vitrification is the direct transformation of water into a “glassy state” without the formation of ice crystals, which avoids the formation of ice crystals inside and outside of the cell through the rapid cooling from the liquid phase to the glassy phase, thus reducing the damage to the callus or cells caused by the ice crystals [13,14]. Vitrification-based cryopreservation refers to the loading and dehydration of the material after pretreatment with a high concentration of a vitrification solution, and the high concentration of the vitrification solution in LN can rapidly form a vitrified state [15].

In nature, only a few plants are capable of vitrification naturally [16] because they lack vitrification-inducing substances or have a high water content. Therefore, cryoprotectants used for vitrification should have at least three main capabilities: a high glass-forming ability, dehydration strength on a colligative basis to dehydrate plant cells to induce a vitrification state, and a cryoprotectant concentration that does not result in excessive toxicity to the plants [17]. Plant vitrification solution (PVS), a commonly used cryoprotectant in vitrification-based cryopreservation, can replace cellular water, alter the freezing behavior, and prevent ultra-water loss [18]. Vitrification-based cryopreservation is widely used in plant germplasm preservation because of its wide adaptability, high freezing rate, and simple operation [19,20], such as for *Solanum tuberosum* [21], *Helianthus tuberosus* [22], *Cynara scolymus* [23], *Chlorophytum borivilianum* [4], and *Smallanthus sonchifolius* [24]. Over the past two decades, the vitrification-based cryopreservation of *A. andraeanum* using suspension cells, axillary buds, and shootlets as materials has been developed [25,26,27]. These three types of explants need to go through more steps after cryopreservation before they can be used for seedling regeneration or as biological breeding materials. In contrast, as an important carrier of genetic resources, the callus with cell totipotency can obtain a large number of regenerated seedlings through induction and can be assisted in breeding new varieties by biological breeding methods in genetic transformation experiments. The callus has a higher value for long-term preservation. As the main material for the breeding, propagation, and production of *A. andraeanum*, the callus has a simple culture technology and low cost. There is a gap in the research on the cryopreservation of *A. andraeanum* callus; its cryopreservation system is still blank.

In this study, for the first time, the key influencing factors of the vitrification-based cryopreservation of the callus of *A. andraeanum* were systematically studied, and a cryopreservation system was established by using exogenous substances. The establishment of vitrification-based cryopreservation has led to a new option for the preservation of *A. andraeanum* germplasm resources and also provides an important theoretical basis for the preservation and utilization of *A. andraeanum* germplasm resources.

## 2. Results

### 2.1. Establishment of Cryopreservation for A. andraeanum

By comparing the levels of various factors in the vitrification-based cryopreservation of herbaceous plants, we analyzed the key steps, including the selection of the sucrose concentration in the preculture, the preculture time, the loading time, and the dehydration time, and the treatments of cryopreservation were established through an orthogonal experimental design. The results were analyzed by nine treatments with an orthogonal design (Figure 1), which showed that there were significant differences in the survival rate of *A. andraeanum* callus after cryopreservation under different treatments. Of the nine treatments, treatment 7 had the highest rate, reaching 48.7%, while treatment 4 had the lowest rate, which was only 10.9%. Compared with treatment 7, the shorter loading time and longer dehydration time of treatment 4 caused the cells to be over-dehydrated without adequate pretreatment, resulting in poor cell conditions before freezing and the inability to withstand subsequent freeze–thaw and dilution treatments.

For the same days of preculture, the higher the sucrose content, the higher the cell viability, and this made the effect of the hypertonic preculture more effective. In the preculture with a sucrose concentration of 0.5 M, the relative cell survival rate of the callus was higher than that in the other two treatments, indicating that this sucrose concentration was more suitable for the cryopreservation of *A. andraeanum*. The cell survival rate in the preculture with a sucrose concentration of 0.3 M was lower than that with 0 M, showing that the degree of damage to the cells caused by cryopreservation in the preculture with a sucrose concentration of 0.3 M was higher than that in the other two treatments, which is not favorable to the long-term preservation of the callus. Among the four influencing factors, namely, the sucrose concentration in the preculture, the preculture time, the loading time, and the dehydration time, the sucrose concentration in the preculture had the greatest influence on the relative cell survival rate, and the preculture time caused the least change in the survival rate.

By analyzing the nine treatments with an orthogonal design, a theoretically optimal treatment was obtained and recorded as treatment 10, i.e., preculture with 0.5 M sucrose for 2 d, a loading time of 60 min, and a dehydration time of 60 min at 25 °C. The relative cell survival rate of treatment 10 was 34.28%, which was a decrease compared with the preculture for 1 d in treatment 7. In summary, during the establishment of cryopreservation for *A. andraeanum* callus, the highest cell survival rate was obtained under treatment 7, which caused the least degree of damage to the cells.

### 2.2. Optimization of Cryopreservation for A. andraeanum

The preculture temperature was changed from 25 °C to 5 °C in treatments 7 and 10. The relative cell survival rates obtained were 10.6% and 13.7%, respectively (Figure 2A). There were significant differences in the relative cell survival rates at different preculture temperatures in both treatments 7 and 10. The cell survival rates of both treatments at a preculture temperature of 5 °C were significantly lower than those at a preculture temperature of 25 °C. Therefore, the callus of *A. andraeanum* was more damaged and intolerant at low temperatures.

In the loading step, the loading solution was changed to 50% PVS2 (LS2) in treatments 7 and 10, and the relative cell survival rates obtained were 52.0% and 55.6%, respectively (Figure 2B). The relative cell survival rate with LS2 was higher than that with the loading solution 1 (LS1) in both treatment 7 and treatment 10, with increases of 6.5% and 38.3%, respectively. There was a significant difference in the relative cell survival rate between LS1 and LS2 in treatment 10. The loading was mainly for bridging the degree of stress between preculture and dehydration. Changing LS1 to LS2, i.e., to 50% PVS, could better adapt the cells to hypertonic stress during dehydration with 100% PVS2. Compared with the components of LS1, LS2 had ethylene glycol and Me_2_SO, both of which have dehydration and osmotic protection properties. In particular, Me_2_SO can lower the freezing point of cells, reduce the formation of ice crystals, and change the permeability of biofilms. The addition of these two components helps cells to better adapt to osmotic dehydration and form a glass state in liquid nitrogen to inhibit the formation of ice crystals, thereby protecting the cells. Therefore, the cell damage of the *A. andraeanum* callus in treatment 10 was significantly reduced with LS2, which was more conducive to the survival of the cells under this condition. In general, changing the loading solution could have played an optimizing role in the cryopreservation system of *A. andraeanum*, and the survival rate of the cells was increased.

Different types of exogenous substances, including antioxidants, abiotic nanomaterials, and plant anti-stress proteins, were added to the PVS2 in the dehydration step. The relative cell survival rates were increased by adding 0.08 mM reduced glutathione (GSH) at 64.6%, 1 mM ascorbic acid (AsA) at 38.4%, 0.1 g L^−1^ single-walled carbon nanotubes (CNTs) at 35.8%, 0.3 g L^−1^ fullerene C_60_ (C60) at 54.5%, and 0.5 μM dehydrin proteins SK_3_ at 34.2% and Y_2_SK_2_ at 51.8% (Figure 3). Therefore, the cell survival rates of *A. andraeanum* callus were significantly different by adding different exogenous substances. The cell survival rate with the addition of 0.08 mM reduced glutathione had the highest survival rate compared with the other five exogenous substances. The addition of 0.3 g L^−1^ fullerene C_60_ or 0.5 μM dehydrin protein Y_2_SK_2_ showed no significant difference in the cell survival rate from CK without adding exogenous substances (treatment 10 with LS2), which indicates that the optimization effect of these two exogenous substances was not obvious. However, the addition of 1 mM ascorbic acid, 0.1 g L^−1^ single-walled carbon nanotubes, and 0.5 μM dehydrin protein SK_3_ significantly reduced the cell survival rate compared with CK, indicating that these three exogenous substances could not only not have an optimization effect but could also aggravate the damage to the cells. In conclusion, the addition of an exogenous substance such as 0.08 mM reduced glutathione could optimize the cryopreservation of *A. andraeanum*.

### 2.3. Standardized Protocol

The procedure for the vitrification-based cryopreservation of *A. andraeanum* callus was established as follows:*A. andraeanum* callus was precultured in 1/2 MS + agar + 0.5 M sucrose + 1 mg L^−1^ thidiazuron (TDZ) + 1 mg L^−1^ 2,4-D solid medium for 2 d at 25 °C.The callus was loaded in 50% PVS2 for 60 min at 25 °C.Dehydration was performed for 60 min at 0 °C in PVS2 (1/2 MS + 0.4 M sucrose + 30% glycerol *w*/*v* + 15% ethylene glycol *w*/*v* + 15% Me_2_SO *w*/*v*) containing 0.08 mM reduced glutathione.The callus was frozen in LN for 1 h and then rapidly thawed at 40 °C for 90 s and washed with an unloading solution (1/2 MS + 1.2 M sucrose) at 25 °C three times, each washing lasting for 10 min.After unloading, the callus was put into a dark culture in a basic medium (1/2 MS + 0.5 M sucrose + agar + 1 mg L^−1^ TDZ + 1 mg L^−1^ 2,4-D) at 25 °C for 1 d. The relative cell survival rate was 64.6%. After 10 days of recovery culture (Figure 4), the cell viability was 32.8%, which ensured the regeneration rate of the callus.

## 3. Discussion

Vitrification-based cryopreservation is currently the widest method for the long-term preservation of plant germplasm resources [11]. This technique has been developed for shoot apices, somatic cultures, cell suspension cultures, and calluses [23]. The effect of vitrification-based cryopreservation is affected by multiple factors, such as the preculture concentration, the number of days of the preculture, the preculture temperature, the loading time and the type of the loading solution, the dehydration time, exogenous substances, and so on. Although it is generally believed that the steps of preculture, loading, and dehydration affect the effect of cryopreservation within a certain range, the different materials of each species need to be determined through experiments to determine the final cryopreservation procedure that can efficiently preserve the materials, and a multi-factor and multi-level orthogonal design experiment is a more efficient research method to establish the cryopreservation procedure [28]. At present, there are only cryopreservation procedures for suspension cells [25], axillary buds [26], and shootlets [27] for *A. andraeanum*, and these three types of cryopreserved material are inferior to calluses in terms of material conditions and seedling formation efficiency. As an important carrier of genetic resources, the callus plays an important role in the conservation, rapid propagation, and biological breeding of germplasm resources, but there is no systematic cryopreservation procedure for calluses. Due to the large differences in the states of the materials, the common cryopreservation procedure cannot be used for the cryopreservation of calluses. In this study, the establishment of vitrification-based cryopreservation has led to a new option for the preservation of *A. andraeanum* germplasm resources, which ensures the stability and consistency of the genetic material of *A. andraeanum* on the basis of reducing the consumption of manpower and material resources.

The orthogonal experimental design method is a statistical technique used to determine the impacts of multiple factors on the experimental results and their optimal combinations. By comparing the levels of various factors in the healing tissue vitrification-based cryopreservation of herbaceous plants, such as *Dioscorea bulbifera* (preculture with 0.3 M sucrose for 1 d, loading for 20 min by loading solution 1, and dehydration for 40 min) [29], *Colocasia esculenta* (preculture with 0.5 M sucrose for 3 d, loading for 20 min by loading solution 1, and dehydration for 40 min) [30], *Agapanthus praecox* (preculture with 0.5 M sucrose for 2 d, loading for 60 min by 60% PVS2, and dehydration for 40 min) [31], *Agrostis stolonifera* (preculture with 0.3 M sucrose for 5 d, loading for 10 min by loading solution 1, and dehydration for 50 min) [32], *Solanum tuberosum* (loading for 20–30 min by 60% PVS2 and dehydration for 50 min) [33], and *Cyclamen persicum* (preculture with 4% sucrose for 3 d and dehydration for 30 min) [34], we analyzed the key steps including the selection of the sucrose concentration in the preculture, the preculture time, the loading time, and the dehydration time for the above-mentioned materials. The optimal treatment for vitrification-based cryopreservation was established through the orthogonal experimental design, which is a special kind of matrix that ensures each level of every factor can be combined with each level of every other factor in a balanced manner, thus reducing the number of necessary experiments.

Preculture is an effective method to improve the physiological state of plant materials and increase the survival rate after cryopreservation. Preculture is very important in the process of cryopreservation, especially for many species that are sensitive to low temperatures or have a high water content. The water content of calluses is higher than that of seeds and embryonic calluses, so the preculture is needed emphatically for calluses to be studied. The resistance of plant materials to osmosis and the increase in the cell survival rate can be effectively improved in the preculture process by culturing the callus in a high-sugar medium. Most of the sugar added to the medium is sucrose, which can better protect the membrane lipid structure of the cells and make the proteins more stable under low-temperature conditions [35]. The effect of cryopreservation was superior when the sucrose concentration of the preculture medium was 0.3–0.7 M and the incubation time was 1–5 d in [36], which is consistent with the results obtained in this study. The relative cell survival rate of calluses was higher under culture with a high sucrose concentration than that with a low concentration. For the same number of days, the higher the sucrose content, and the higher the cell viability. In the preculture with sucrose at low concentrations, the cell survival rate was higher with longer days, while in the preculture with high sucrose concentrations, the cell survival rate was higher with a shorter number of days.

Among the nine groups of experiments in the orthogonal design, treatment 7 had the highest cell survival rate, and the theoretically optimal treatment was treatment 10. The difference between the two was only in the number of days of preculture, where treatment 7 was 1 day, and treatment 10 was 2 days. From the experimental results of treatments 7 and 10, it can be concluded that under the conditions of the same preculture concentration, loading time, and dehydration time, the number of days of preculture had an influence on the cell survival rate after the vitrification-based cryopreservation of *A. andraeanum* callus, i.e., hypertonic preculture is required for the cryopreservation of *A. andraeanum* callus. Since *A. andraeanum* is adapted to high temperatures and humidity, this experiment showed a significant decrease in cell survival by changing the preculture temperature from 25 °C to 5 °C, indicating that the callus of *A. andraeanum* is also not resistant to low temperatures. This is consistent with the statement that the origin of *A. andraeanum* is tropical rainforest areas such as Costa Rica and Colombia. Therefore, the callus of *A. andraeanum* can only be hypertonic but not low-temperature-tolerant when it is precultured.

There is usually a loading process for plant material before dehydration from the vitrification solution. A mixture of cryoprotectants is applied to reduce the water content of the tissue and avoid damage to the material due to drastic changes in osmotic pressure [37]. In vitrification cryopreservation, the loading process is mainly 10–60 min, but for some plant materials, this step may not be necessary [38,39]. In this study, a longer loading treatment combined with an appropriate period of dehydration treatment resulted in high cell viability. Peng [33] used 60% PVS2 as a loading solution to deal with potato callus, which resulted in 57.5% cell survival after cryopreservation. Under the condition that the loading time remained unchanged, the relative cell survival rate of the callus was improved by adjusting the loading solution to 50% PVS2, which indicated that the loading step with 50% PVS2 was more conducive to the cell survival of *A. andraeanum*. Furthermore, the toxicity of the loading treatment to the cells decreased with the growth in the cultivation time. Whether changing the temperature of the preculture or the formulation of the loading solution, the cell survival rate of treatment 10 was higher than that of treatment 7, so the theoretically optimal system was analyzed by the orthogonal design, and the subsequent optimized treatments all used treatment 10, and the loading solution was changed to 50% PVS2.

The most critical step of the vitrification cryopreservation of plants is dehydration, and the most important factor is to strictly control the dehydration time of the plant material. The dehydration time depends on the characteristics of the material, including the genotype, frost resistance, age, and physiological state of the cells [13]. Due to the high concentration of Me_2_SO and sucrose in the vitrification solution, a short treatment time may not allow the free water to completely penetrate into the cells, resulting in the production of ice crystals during rapid cooling. If the treatment time is too long, ionic poison and excessive dehydration will be produced, which will directly cause reduced cell viability and even death. According to the existing studies, the dehydration times of different materials vary greatly, with the dehydration time of the zygotic embryo of *Bletila striata* being 3 h [40] and the dehydration time of the embryonic callus of *Picea mariana* being 30 min [38]. According to the results of this study, the callus was treated in the vitrification solution for 60 min to achieve a high cell survival rate.

In this study, different types of exogenous substances were added in a unifactorial manner to the PVS2, including antioxidants, abiotic nanomaterials, and dehydrin proteins. Among antioxidants, GSH is more effective than AsA. In the cryopreservation of *Agapanthus praecox* embryogenic callus, the relative cell viability after GSH addition during the dehydration step was higher than that in the ASA-supplemented treatment group [41]. This is the same as the results of this study. Nanomaterials have a good effect on improving cryoprotectants due to their small particle size and large specific surface [42]. This study found that adding fullerene C_60_ was better than adding single-walled carbon nanotubes. This result was also found in a previous study [43]. Two dehydrins from *Agapanthus praecox* (Y_2_SK_2_ and SK_3_) showed important protective effects under complex stresses [44]. This study used these two dehydrins as exogenous substances and found that the addition of Y_2_SK_2_ had higher cell viability than the addition of SK_3_. In previous studies, adding Y_2_SK_2_ and SK_3_ to the plant vitrification solution increased the survival ratio of wild-type *Arabidopsis thaliana* seedlings from 24% to 50% and 46%, respectively [44]. These two dehydrins, as exogenous substances, can also improve the cell viability of *Arabidopsis thaliana* [45] and *Agapanthus praecox* [46].

Among all these exogenous substances, the cell survival rate obtained from the dehydration treatment by PVS2 containing 0.08 mM GSH was as high as 64.6%. In previous studies, the main factor limiting the relative survival of cells after cryopreservation is reactive oxygen species-induced oxidative stress; nevertheless, antioxidant systems, including catalase, peroxisome, superoxide dismutase, ascorbic acid, and reduced glutathione, can effectively alleviate the state of intracellular oxidative stress [47]. GSH is regarded as one of the most important cellular antioxidants, and it participates in the AsA-GSH cycle in plants to reduce the accumulation of reactive oxygen species and maintain the homeostasis of intracellular oxidation, increasing the activity of cellular antioxidant enzymes and antioxidants [48]. The relative survival of cells could be as high as 88.3% by adding GSH to PVS2 in the cryopreservation of *Agapanthus praecox* embryogenic callus [48]. Therefore, the addition of GSH as an exogenous substance to PVS2 could improve the relative cell survival rate after cryopreservation of *A. andraeanum* callus, which is in agreement with the previous research results.

## 4. Materials and Methods

### 4.1. Plant Materials

The *A. andraeanum* variety ‘Pink Champion’ was used as the experimental material in this study. The calluses were induced from young leaves on 1/2 MS medium [49] (Sangon Biotech, Shanghai, China) supplemented with sucrose (Sangon Biotech, Shanghai, China), agar (Sangon Biotech, Shanghai, China), 1 mg L^−1^ TDZ (Sangon Biotech, Shanghai, China), and 1 mg L^−1^ 2,4-D (Sangon Biotech, Shanghai, China) at 25 °C in the dark by subculturing monthly. This was the group with the highest callus induction rate among the following five hormone ratios: 0.4 mg L^−1^ TDZ + 0.5 mg L^−1^ 2,4-D; 0.4 mg L^−1^ TDZ + 1 mg L^−1^ 2,4-D; 1 mg L^−1^ TDZ + 0.5 mg L^−1^ 2,4-D; 1 mg L^−1^ TDZ + 1 mg L^−1^ 2,4-D; and 0.4 mg L^−1^ TDZ (unpublished). After 14 days of subculture, 0.2 g of callus was used as the replication for subsequent experiments (Figure 5).

### 4.2. Cryopreservation Procedure

The key point of vitrification-based cryopreservation is the vitrification state before putting in liquid nitrogen. In the related studies on the cryopreservation of herbaceous plants [29,30,31,32,33,34], we analyzed the key steps, including the selection of the sucrose concentration in the preculture, the preculture time, the loading time, and the dehydration time, and the optimal treatment of vitrification-based cryopreservation was established through the orthogonal experimental design (Table 1). Each procedure was repeated 3 times.

An amount of 0.2 g of callus was selected and inoculated onto 1/2 MS solid medium containing different sucrose concentrations (0, 0.3, or 0.5 M; 0 represents no preculture) and incubated in dark culture (1, 2, or 3 d) at 25 °C. An amount of 0.2 g of callus was placed into 2 mL cryovials with 1.8 mL of loading solution 1 (LS1) [1/2 MS + 0.4 M sucrose + 2 M glycerol (SINOPHARM, Beijing, China)] and processed at room temperature (20, 40, or 60 min). LS1 was replaced with 1.8 mL of vitrification solution PVS2 [1/2 MS + 0.4 M sucrose + 30% glycerol *w*/*v* + 15% ethylene glycol *w*/*v* (SINOPHARM, Beijing, China) + 15% Me_2_SO (SINOPHARM, Beijing, China) *w*/*v*] [48], and dehydration was carried out at 0 °C (30, 60, or 90 min).

The cryovials were rapidly plunged into LN for 1 h and thawed rapidly for 90 s in a 40 °C water bath. The PVS2 was replaced with an unloading solution (1/2 MS + 1.2 M sucrose) for 30 min at room temperature, and the unloading solution was replaced with a fresh one every 10 min. The callus was placed in a basic solid medium (1/2 MS + sucrose + agar + 1 mg L^−1^ TDZ + 1 mg L^−1^ 2,4-D) for 24 h in dark culture and used to determine the relative cell survival.

### 4.3. Optimization of Vitrification Cryopreservation

The following three different systems were adjusted and optimized separately, and the effectiveness of the optimization was judged based on the relative cell survival rate. Each procedure was repeated 3 times.

The preculture temperature was decreased from 25 °C to 5 °C, and the above cryopreservation steps were repeated.LS1 was changed to 50% PVS2 as LS2 (1/2 MS liquid medium: PVS2 = 1:1, *v*/*v*), and the above cryopreservation steps were repeated.Exogenous substances such as abiotic nanomaterials (0.3 g L^−1^ fullerene C_60_ (XFNANO, Nanjing, China) [43], 0.1 g L^−1^ single-walled carbon nanotubes (XFNANO, Nanjing, China) [41,43]), antioxidants (0.08 mM reduced glutathione (Sangon Biotech, Shanghai, China) [41,48] and 1 mM ascorbic acid (Sangon Biotech, Shanghai, China) [41]), plant anti-stress proteins (0.5 μM dehydrin proteins SK_3_ [44,45,46] and Y_2_SK_2_ [44,45,46]) were added to the PVS2, and the above steps of cryopreservation were repeated.

### 4.4. Viability Detection

In order to detect the survival of the cryopreserved callus, the 2,3,5-triphenyltetrazolium chloride (TTC) (Sangon Biotech, Shanghai, China) protocol was used in this study [48]. The callus (0.05 g) after 24 h of recovery and 10 d of recovery was put into 2 mL of 0.8% TTC [dissolved in 50 mM phosphate buffer (pH 7.4)] and kept in the dark for 20 h at 25 °C. The TTC staining solution was removed, the callus was rinsed with sterile water 3 times, 5 mL of 95% ethanol was added, and the callus was water-bathed at 80 °C for 50 min. The callus was centrifuged at 3000 g for 5 min, and the optical density value of the supernatant was tested at 485 nm. The relative absorbance values were used to express the relative cell viability of each treatment after cryopreservation. Each sample procedure was repeated 3 times. The formula is as follows:

Relative survival (%) = (Absorbance value of treated samples/absorbance value of untreated samples) × 100%

### 4.5. Statistical Analysis

The orthogonal experimental design and analysis of the results were performed by Orthogonal Design Assistant V3.1. The orthogonal design was followed to improve the recovery over preliminary optimal treatments. The experiments were repeated three times. One-way ANOVA was used to analyze the differences followed by the least-significant difference multiple-range test using the Statistics Analysis System 9.1.3 software (SAS Institute, Inc., Cary, NC, USA).

## 5. Conclusions

As an important carrier of genetic resources, the long-term preservation of the callus of *A. andraeanum* is extremely important for the production and selection of new varieties. In this study, the key treatments in cryopreservation were systematically screened for the first time, and the procedure was finally established through optimization experiments using the orthogonal experimental method of four factors and three levels in the sucrose concentration of the preculture, the preculture time, the loading time, and the dehydration time. After this, the cryopreservation procedure was optimized by changing the preculture temperature and loading solution and adding exogenous substances to the PVS2, including GSH, AsA, CNTs, C60, SK_3_, and Y_2_SK_2_. The cryopreservation procedure of the callus by vitrification was as follows: the callus was precultured at 25 °C for 2 d; it was loaded in 50% PVS2 at 25 °C for 60 min; it was dehydrated with PVS2 containing 0.08 mM reduced glutathione (GSH) at 0 °C for 60 min. After rapid-cooling in liquid nitrogen for 1 h, it was rapid-warming in a water bath at 40 °C for 90 s and unloaded for 30 min. The cell relative survival rate of the cryopreserved callus was 64.60% after 1 d of recovery. This study provides an important theoretical basis for the preservation and utilization of *A. andraeanum* germplasm resources.

## Figures and Tables

**Figure 1 plants-13-03106-f001:**
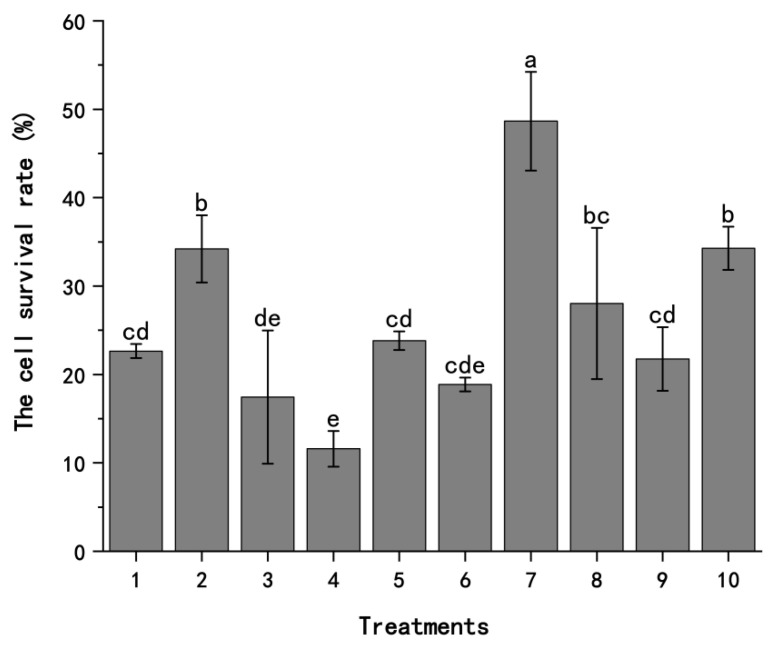
The results of the treatments established by the orthogonal design in the cryopreservation of *A. andraeanum* callus. The treatments refer to differences in the preculture sucrose concentration, preculture time, loading time, and dehydration time. Treatment 1: no preculture, loading for 20 min, and dehydration for 30 min; treatment 2: no preculture, loading for 40 min, and dehydration for 60 min; treatment 3: no preculture, loading for 60 min, and dehydration for 90 min; treatment 4: preculture with 0.3 M sucrose for 1 d, loading for 40 min, and dehydration for 90 min; treatment 5: preculture with 0.3 M sucrose for 2 d, loading for 60 min, and dehydration for 30 min; treatment 6: preculture with 0.3 M sucrose for 3 d, loading for 20 min, and dehydration for 60 min; treatment 7: preculture with 0.5 M sucrose for 1 d, loading for 60 min, and dehydration for 60 min; treatment 8: preculture with 0.5 M sucrose for 2 d, loading for 20 min, and dehydration for 90 min; treatment 9: preculture with 0.5 M sucrose for 3 d, loading for 40 min, and dehydration for 30 min; and treatment 10: preculture with 0.5 M sucrose for 2 d, loading for 60 min, and dehydration for 60 min. Each procedure was repeated 3 times. The percentages are the means of three replicate experiments. The error bars presented are the means ± SE. Letters denote significant differences in the results for each treatment at the 0.05 level, and treatments were conducted as per the orthogonal design.

**Figure 2 plants-13-03106-f002:**
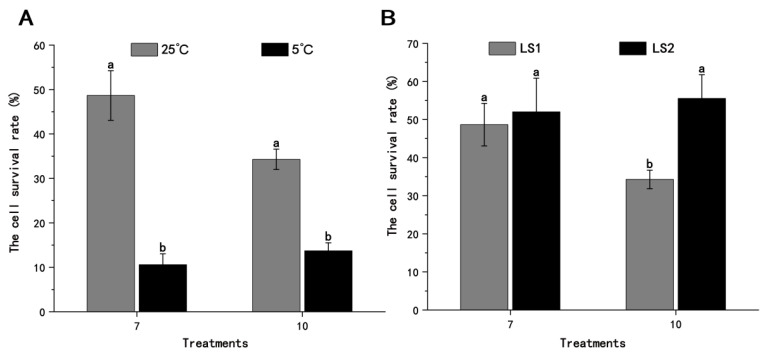
Effects of (**A**) preculture temperature and (**B**) loading solution on the cell survival rate of *A. andraeanum* callus by cryopreservation with treatment 7 (preculture with 0.5 M sucrose for 1 d, loading for 60 min, and dehydration for 60 min) and treatment 10 (preculture with 0.5 M sucrose for 2 d, loading for 60 min, and dehydration for 60 min). The treatments in (**A**) refer to differences in the preculture temperature (25 °C and 5 °C), and the treatments in (**B**) refer to differences in the loading solution (LS1 and LS2). LS1: 1/2 MS + 0.4 M sucrose + 2 M glycerol; LS2: 1/2 MS; liquid medium: PVS2 = 1:1, *v*/*v*. Each procedure was repeated 3 times. The percentages are the means of three replicate experiments. The error bars presented are the means ± SE. Letters denote significant differences in the results for each treatment at the 0.05 level.

**Figure 3 plants-13-03106-f003:**
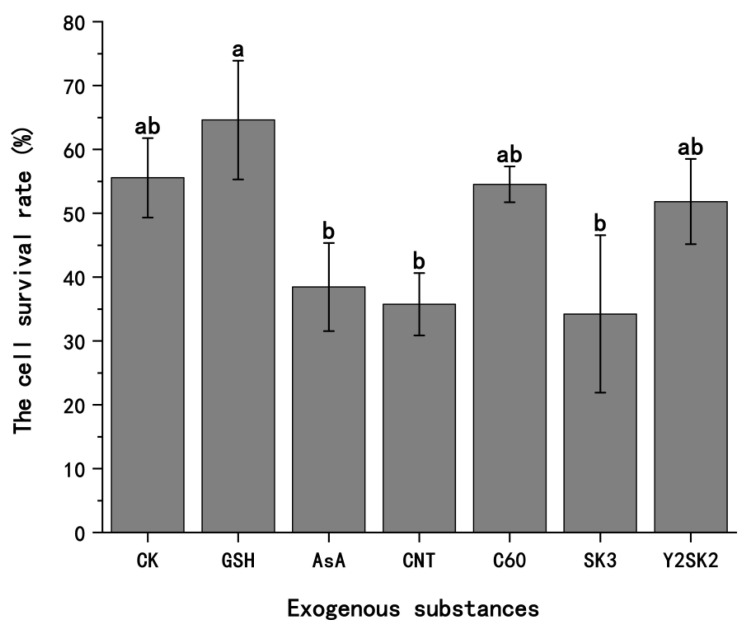
The effects of exogenous substances on survival of cryopreserved *A. andraeanum* callus. The standard condition was CK without adding exogenous substances to PVS2 (treatment 10 with LS2), including 0.08 mM reduced glutathione (GSH); 1 mM ascorbic acid (AsA); 0.1 g L^−1^ single-walled carbon nanotubes (CNTs); 0.3 g L^−1^ fullerene C_60_ (C60); 0.5 μM dehydrin protein SK_3_ (SK3); and 0.5 μM dehydrin protein Y_2_SK_2_ (Y2SK2). Each procedure was repeated 3 times. The percentages are the means of three replicate experiments. The error bars presented are the means ± SE. Letters denote significant differences in the results for each treatment at the 0.05 level.

**Figure 4 plants-13-03106-f004:**
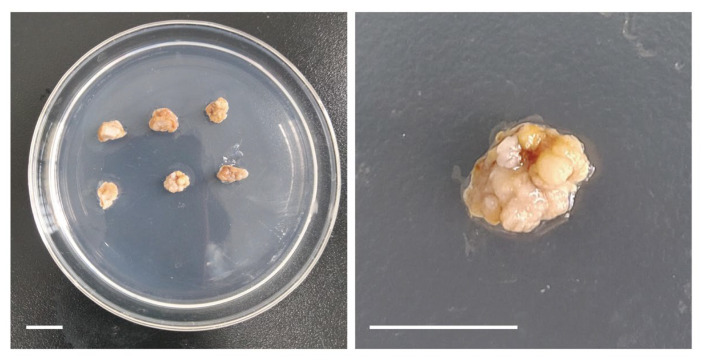
The callus cultured for 10 days after cryopreservation. The cryopreservation procedure was as follows: The callus was precultured in 0.5 M solid sucrose medium for 2 d at 25 °C and loaded in 50% PVS2 for 60 min at 25 °C. Dehydration was performed for 60 min at 0 °C in PVS2 containing 0.08 mM GSH. The callus was frozen in LN for 1h and then rapidly thawed at 40 °C for 90 s and washed with an unloading solution at 25 °C 3 times, each washing lasting for 10 min, and put into a dark culture in a basic medium at 25 °C. Scale bars indicate 1 cm.

**Figure 5 plants-13-03106-f005:**
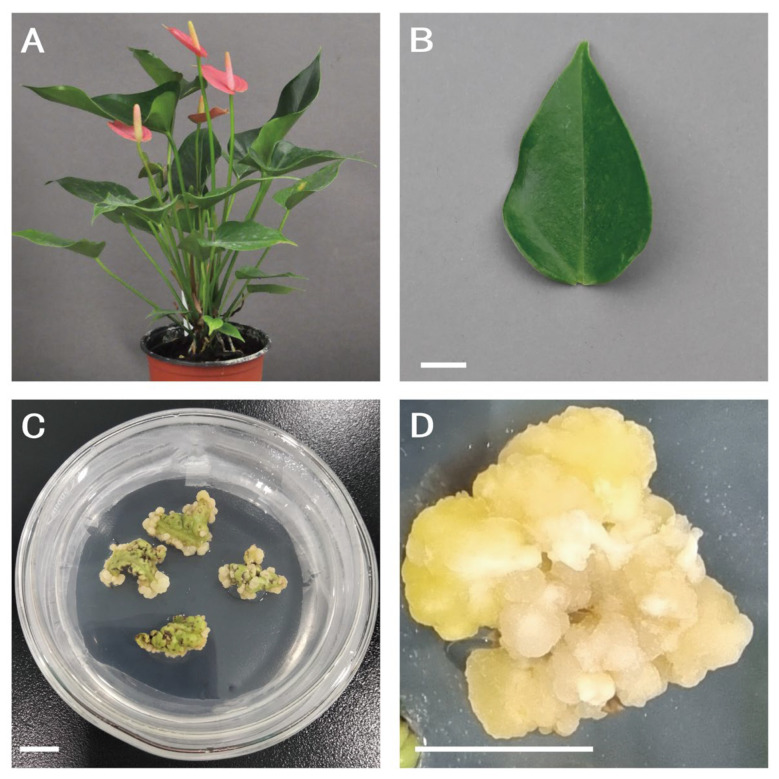
Plant materials for cryopreservation of *A. andraeanum*. (**A**) *A. andraeanum* variety ‘Pink Champion’. (**B**) Young leaf culture for callus induction. (**C**) Callus cultures were initiated from leaf explants. (**D**) An amount of 0.2 g of callus was used as the replication for subsequent experiments after 14 days of subculture. Scale bars indicate 1 cm in (**B**–**D**).

**Table 1 plants-13-03106-t001:** Four-factor, three-level orthogonal experimental design.

No.	Preculture Sucrose Concentration (M)	Preculture Time (d)	Loading Time (min)	Dehydration Time (min)
1	0	1	20	30
2	0	2	40	60
3	0	3	60	90
4	0.3	1	40	90
5	0.3	2	60	30
6	0.3	3	20	60
7	0.5	1	60	60
8	0.5	2	20	90
9	0.5	3	40	30

## Data Availability

Data are contained within the article.

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
