# Peer review of "The Development of a Procedure for the Cryopreservation of the Callus of Anthurium andraeanum by Vitrification"

_plants, 2024, doi:10.3390/plants13213106_

Round 1
Reviewer 1 Report
Comments and Suggestions for Authors
In this study, the authors used callus of Anthurium andraeanum for cryopreservation experiments, developing the callus cultures with a single medium without any detailed characterization and parameters. The authors applied a standard vitrification protocol, testing a few variations in each step of the protocol, and assessed cell survival using a TTC viability test. However, the manuscript does not demonstrate cell or callus growth post-cryopreservation, and it lacks data on survival or regrowth after the process. The study lacks scientific thoroughness, as there is insufficient justification or discussion for the selection and concentration of treatments used. Additionally, there is no evidence of shoot regeneration (organogenesis) after cryopreservation, which limits the practical applicability of the findings.
Comments on the Quality of English LanguageN/A
Author Response
Comments: In this study, the authors used callus of Anthurium andraeanum for cryopreservation experiments, developing the callus cultures with a single medium without any detailed characterization and parameters. The authors applied a standard vitrification protocol, testing a few variations in each step of the protocol, and assessed cell survival using a TTC viability test. However, the manuscript does not demonstrate cell or callus growth post-cryopreservation, and it lacks data on survival or regrowth after the process. The study lacks scientific thoroughness, as there is insufficient justification or discussion for the selection and concentration of treatments used. Additionally, there is no evidence of shoot regeneration (organogenesis) after cryopreservation, which limits the practical applicability of the findings.
Response: Thanks to the reviewer for the comments. The callus-induced medium formula was supplemented, which was the group with the highest callus induction rate among the five hormone ratios. The discussion of the selection and concentration of treatments used has been modified. The recovery growth experiment was supplemented, and the cell viability was analyzed for 10 days after the process and the results were added to the result.
Reviewer 2 Report
Comments and Suggestions for Authors
Dear editor, dear author,
here is my review of the article plants-3152816:-Establishment and Optimization of Vitrification-Based Cryopreservation for Callus of Anthurium andraeanum:
The subject of the manuscript is the development of a vitrification-dehidration procedure for the callus cryopreservation of Anthurium andraeanum.
Title:
The title should be more precise: “Development of a procedure for cryopreservation of callus of Anthurium andraeanum by vitrification-dehydration«.
Abstract: OK
Introduction:
In the first section, the authors present the economic value of this plant and its in vitro culture, citing two master's theses. There are many better and more accessible reports on the internet about the importance of micropropagation and tissue culture of this species. Please replase them with them.
The introduction is too long and deals with cryopreservation in too general a way
It should focus on (1) vitrification-dehydration and (2) cryopreservation of Anthurium.
There is another reference on cryopreservation not mentioned in the article:
Hanan, M. A., & Sherif, S. S. (2013). Conservation Plantlets of Anthurium andraeanum (Lindeu ex Andre) By using in Vitro. Middle East j, 2(2), 36-43.
The sentences between lines 186-203 should be omitted without changing the introduction to this topic and replaced by a brief description of vitrification and vitrification-dehydration.
It should clearly state the purpose and aim of the study!
The sentences from lines 220 and 2023 must be part of the discussion.
Results:
The sentences in lines 226-263 are repetitions of material and method and should be removed from the results. Do not repeat the material and methods and only explain the results.
Figures: Should be self-explanatory.
Mean and SE are described in all figures, but without the number of explants used for the calculation (observations?).
Several details of the statistics are missing, numbers (n=?) of observations are missing.
The authors need to explain how they calculate mean percentages ± SE. Is it even possible to calculate mean percentages?
Percentages and mean values should be written with one or no decimal places.
Percentages are mean values of three repetitions? Explain this in more detail.
See:
https://www.indeed.com/career-advice/career-development/how-to-calculate-average-percentage https://www.robertoreif.com/blog/2018/1/7/why-you-should-be-careful-when-averaging-percentages https://sciencing.com/calculate-weighted-average-5328019.html
Is it possible to do a (full name) test (LSD) with percentages? A I know it is not possible! If it is possible, please explain when and how!
ANOVA calculation with percentages is also a wrong calculation as far as I know!
LDS test – is a test for normally distributed data. Are the data normally distributed? Are the variances significantly different?
Number of spores or gametophytes examined – 5 replicates with how many spores? How many replicates?
Statistics:
Normality of the data - - Is ANOVA allowed if the data are normally distributed!
What was compared?
Mean value of percentages – this must not be calculated – this is wrong!!!!!! -
ANOVA calculation with percentages is also wrong as far as I know!
LDS test – is a test for normally distributed data. Are the data normally distributed? Are the variances significantly different?
How many repetitions?
Material and methods:
Please explain what you think of each of the cryo methods listed.
The reference for PVS2 is missing.
The experimental design is missing what was meant by callus – one callus, two calli, …....what was used for calculation?
What references was the optimization based on?
Statistics:
The authors need to explain how they calculate mean percentages ± SE. Is it even possible to calculate mean percentages?
See:
https://www.indeed.com/career-advice/career-development/how-to-calculate-average-percentage https://www.robertoreif.com/blog/2018/1/7/why-you-should-be-careful-when-averaging-percentages https://sciencing.com/calculate-weighted-average-5328019.html
Is it possible to do a (full name) test (LSD) with percentages? A I know it is not possible! If it is possible, please explain when and how!
Percentages and averages should be written with one or no decimal places.
Discussion:
As an introduction, the discussion is too long and too general and discuss cryopreservation at some parts general - and this is boring.
It should focus on (1) vitrification-dehydration and on (2) cryopreservation of Anthurium.
There is another reference on cryopreservation not mentioned in the article:
Hanan, M. A., & Sherif, S. S. (2013). Conservation Plantlets of Anthurium andraeanum (Lindeu ex Andre) By using in Vitro. Middle East j, 2(2), 36-43.
There is no conclusion!
Yours sincerely, Reviewer
Author Response
Thanks to the reviewer for the comments. We give the point-by-point response to the reviewer’s comments as below.
Title:
Comments 1: The title should be more precise: “Development of a procedure for cryopreservation of callus of Anthurium andraeanum by vitrification-dehydration«.
Response 1: We have revised the title based on the reviewers' comment.
Abstract: OK
Introduction:
Comments 2: In the first section, the authors present the economic value of this plant and its in vitro culture, citing two master's theses. There are many better and more accessible reports on the internet about the importance of micropropagation and tissue culture of this species. Please replase them with them.
Response 2: We have replaced them.
Comments 3: The introduction is too long and deals with cryopreservation in too general a way
It should focus on (1) vitrification-dehydration and (2) cryopreservation of Anthurium.
Response 3: The introduction has been streamlined and focuses on vitrification and cryopreservation of Anthurium.
Comments 4: There is another reference on cryopreservation not mentioned in the article:
Hanan, M. A., & Sherif, S. S. (2013). Conservation Plantlets of Anthurium andraeanum (Lindeu ex Andre) By using in Vitro. Middle East j, 2(2), 36-43.
Response 4: We have added this reference.
Comments 5: The sentences between lines 186-203 should be omitted without changing the introduction to this topic and replaced by a brief description of vitrification and vitrification-dehydration.
Response 5: The introduction has been streamlined and especially these lines.
Comments 6: It should clearly state the purpose and aim of the study!
Response 6: We have revised and refined the purpose and aim of the study.
Comments 7: The sentences from lines 220 and 2023 must be part of the discussion.
Response 7: These sentences have already been put in the discussion.
Results:
Comments 8: The sentences in lines 226-263 are repetitions of material and method and should be removed from the results. Do not repeat the material and methods and only explain the results.
Response 8: It is recommended that it be appropriately retained here, as the material and method are in the second half of the manuscript. This part facilitates the reader's comprehension of the result.
Comments 9: Figures: Should be self-explanatory.
Response 9: Figure legends have been improved.
Comments 10: Mean and SE are described in all figures, but without the number of explants used for the calculation (observations?).
Response 10: Each procedure was repeated 3 times. We have added the number of explants used for the calculation.
Comments 11: Several details of the statistics are missing, numbers (n=?) of observations are missing.
Response 11: Each procedure was repeated 3 times. We have added the number of explants used for the calculation.
Comments 12: The authors need to explain how they calculate mean percentages ± SE. Is it even possible to calculate mean percentages?
Response 12: Referring to the existing studies on cryopreservation, it can be performed by regrowth rate and cell viability including mean percentages ± SE. In this study, each treatment was divided into triplicate replicates and experiments were performed in three separate cryovials. Three replicates of cell viability data can be obtained, and the mean and standard deviation were further calculated, as well as the significance of the difference analysis.
References:
Chen, G.Q.; Ren, L.; Zhang, D.; Shen, X.H. Glutathione improves survival of cryopreserved embryogenic calli of Agapanthus praecox subsp. orientalis. Acta Physiol. Plant. 2016, 38, 250. (Figure 2)
Yang, Z.; Sheng, J.Y.; Lv, K.; Ren, L.; Zhang, D. Y2SK2 and SK3 type dehydrins from Agapanthus praecox can improve plant stress tolerance and act as multifunctional protectants. Plant Sci. 2019, 284, 143–160. (Figure 15)
Ren, L.; Deng, S.; Chu, Y.; Zhang, Y.; Zhao, H.; Chen, H.; Zhang, D. Single-wall carbon nanotubes improve cell survival rate and reduce oxidative injury in cryopreservation of Agapanthus praecox embryogenic callus. Plant Methods 2020, 16, 130. (Figure 1)
Zhang, D.; Yang, T.C.; Ren, L. Y2SK2- and SK3-type dehydrins from Agapanthus praecox act as protectants to improve plant cell viability during cryopreservation. Plant Cell Tiss. Org. 2021, 144, 271–279. (Figure 2)
Arora, R. Exploring freeze-injury mechanism through ion-specific analysis of leachate from reversibly versus irreversibly injured spinach (Spinacia oleracea L.) leaves. Cryobiology, 2024, 117, 104954. (Figure 1)
Lee, H.; Park, H.; Park, S.-U.; Kim, H. Liquid overlay-induced donor plant vigor and initial ammonium-free regrowth medium are critical to the cryopreservation of Scrophularia kakudensis. Plants 2024, 13, 2408. (Figure 5)
Wasileńczyk, U.; Wawrzyniak, M.K.; Martins, J.P.R.; Paulina Kosek P.; Chmielarz P. Cryopreservation of sessile oak (Quercus petraea (Matt.) Liebl.) plumules using aluminium cryo-plates: influence of cryoprotection and drying. Plant Methods 2024, 20, 53. (Figure 2)
Wang, M.R.; Bao, J.H.; Ma, X.Y.; Yan, Z.H.; Cui, Z.H.; Zhu, L.Y.; Zhang, D.; Wang, Q.C. Vitrification cryo-foil method for shoot tip cryopreservation and virus eradication in apple. Cryobiology, 2024, 117, 104957. (Figure 1)
Comments 13: Percentages and mean values should be written with one or no decimal places.
Response 13: We have modified them.
Comments 14: Percentages are mean values of three repetitions? Explain this in more detail.
Response 14: In this study, each treatment was divided into triplicate replicates and experiments were performed in three separate cryovials. Three replicates of cell viability data can be obtained, and the mean and standard deviation were further calculated.
See:
https://www.indeed.com/career-advice/career-development/how-to-calculate-average-percentage https://www.robertoreif.com/blog/2018/1/7/why-you-should-be-careful-when-averaging-percentages https://sciencing.com/calculate-weighted-average-5328019.html
Comments 15: Is it possible to do a (full name) test (LSD) with percentages? A I know it is not possible! If it is possible, please explain when and how!
ANOVA calculation with percentages is also a wrong calculation as far as I know!
Response 15: Referring to the existing studies on cryopreservation, they used the one-way ANOVA to analyze differences followed by the least significant difference multiple range test. In this study, each treatment was divided into triplicate replicates and experiments were performed in three separate cryovials. Three replicates of cell viability data can be obtained, and we turn the percentages into decimals. Three replicates of cell viability data can be further calculated, as well as the significance of the difference analysis.
References:
Chen, G.Q.; Ren, L.; Zhang, D.; Shen, X.H. Glutathione improves survival of cryopreserved embryogenic calli of Agapanthus praecox subsp. orientalis. Acta Physiol. Plant. 2016, 38, 250.
Yang, Z.; Sheng, J.Y.; Lv, K.; Ren, L.; Zhang, D. Y2SK2 and SK3 type dehydrins from Agapanthus praecox can improve plant stress tolerance and act as multifunctional protectants. Plant Sci. 2019, 284, 143–160.
Ren, L.; Deng, S.; Chu, Y.; Zhang, Y.; Zhao, H.; Chen, H.; Zhang, D. Single-wall carbon nanotubes improve cell survival rate and reduce oxidative injury in cryopreservation of Agapanthus praecox embryogenic callus. Plant Methods 2020, 16, 130.
Zhang, D.; Yang, T.C.; Ren, L. Y2SK2- and SK3-type dehydrins from Agapanthus praecox act as protectants to improve plant cell viability during cryopreservation. Plant Cell Tiss. Org. 2021, 144, 271–279.
Lee, H.; Park, H.; Park, S.-U.; Kim, H. Liquid overlay-induced donor plant vigor and initial ammonium-free regrowth medium are critical to the cryopreservation of Scrophularia kakudensis. Plants 2024, 13, 2408.
Comments 16: LDS test – is a test for normally distributed data. Are the data normally distributed? Are the variances significantly different?
Response 16: According to our results, the data are normally distributed and the variances significantly different.
Comments 17: Number of spores or gametophytes examined – 5 replicates with how many spores? How many replicates?
Response 17: Each procedure was repeated 3 times. We have added the number of explants used for the calculation.
Statistics:
Comments 18: Normality of the data - - Is ANOVA allowed if the data are normally distributed!
What was compared?
Response 18: Referring to the existing studies on cryopreservation, they used the one-way ANOVA to analyze differences followed by the least significant difference multiple range test. In this study, each treatment was divided into triplicate replicates and experiments were performed in three separate cryovials. Three replicates of cell viability data can be obtained, and we turn the percentages into decimals. Three replicates of cell viability data can be further calculated, as well as the significance of the difference analysis.
References:
Chen, G.Q.; Ren, L.; Zhang, D.; Shen, X.H. Glutathione improves survival of cryopreserved embryogenic calli of Agapanthus praecox subsp. orientalis. Acta Physiol. Plant. 2016, 38, 250.
Yang, Z.; Sheng, J.Y.; Lv, K.; Ren, L.; Zhang, D. Y2SK2 and SK3 type dehydrins from Agapanthus praecox can improve plant stress tolerance and act as multifunctional protectants. Plant Sci. 2019, 284, 143–160.
Ren, L.; Deng, S.; Chu, Y.; Zhang, Y.; Zhao, H.; Chen, H.; Zhang, D. Single-wall carbon nanotubes improve cell survival rate and reduce oxidative injury in cryopreservation of Agapanthus praecox embryogenic callus. Plant Methods 2020, 16, 130.
Zhang, D.; Yang, T.C.; Ren, L. Y2SK2- and SK3-type dehydrins from Agapanthus praecox act as protectants to improve plant cell viability during cryopreservation. Plant Cell Tiss. Org. 2021, 144, 271–279.
Lee, H.; Park, H.; Park, S.-U.; Kim, H. Liquid overlay-induced donor plant vigor and initial ammonium-free regrowth medium are critical to the cryopreservation of Scrophularia kakudensis. Plants 2024, 13, 2408.
Comments 19: Mean value of percentages – this must not be calculated – this is wrong!!!!!! -
Response 19: In this study, each treatment was divided into triplicate replicates and experiments were performed in three separate cryovials. Three replicates of cell viability data can be obtained, and we turn the percentages into decimals to further calculate mean value of percentages.
Comments 20: ANOVA calculation with percentages is also wrong as far as I know!
Response 20: Referring to the existing studies on cryopreservation, they used the one-way ANOVA to analyze differences followed by the least significant difference multiple range test. In this study, each treatment was divided into triplicate replicates and experiments were performed in three separate cryovials. Three replicates of cell viability data can be obtained, and we turn the percentages into decimals. Three replicates of cell viability data can be further calculated, as well as the significance of the difference analysis.
References:
Chen, G.Q.; Ren, L.; Zhang, D.; Shen, X.H. Glutathione improves survival of cryopreserved embryogenic calli of Agapanthus praecox subsp. orientalis. Acta Physiol. Plant. 2016, 38, 250.
Yang, Z.; Sheng, J.Y.; Lv, K.; Ren, L.; Zhang, D. Y2SK2 and SK3 type dehydrins from Agapanthus praecox can improve plant stress tolerance and act as multifunctional protectants. Plant Sci. 2019, 284, 143–160.
Ren, L.; Deng, S.; Chu, Y.; Zhang, Y.; Zhao, H.; Chen, H.; Zhang, D. Single-wall carbon nanotubes improve cell survival rate and reduce oxidative injury in cryopreservation of Agapanthus praecox embryogenic callus. Plant Methods 2020, 16, 130.
Zhang, D.; Yang, T.C.; Ren, L. Y2SK2- and SK3-type dehydrins from Agapanthus praecox act as protectants to improve plant cell viability during cryopreservation. Plant Cell Tiss. Org. 2021, 144, 271–279.
Lee, H.; Park, H.; Park, S.-U.; Kim, H. Liquid overlay-induced donor plant vigor and initial ammonium-free regrowth medium are critical to the cryopreservation of Scrophularia kakudensis. Plants 2024, 13, 2408.
Comments 21: LDS test – is a test for normally distributed data. Are the data normally distributed? Are the variances significantly different?
Response 21: According to our results, the data are normally distributed and the variances significantly different.
Comments 22: How many repetitions?
Response 22: Each procedure was repeated 3 times. We have added the number of explants used for the calculation.
Material and methods:
Comments 23: Please explain what you think of each of the cryo methods listed.
Response 23: We have explained each of the cryo methods listed in Materials and Methods.
Comments 24: The reference for PVS2 is missing.
Response 24: We have added the reference.
Comments 25: The experimental design is missing what was meant by callus – one callus, two calli, …....what was used for calculation?
Response 25: We have added "0.2 g of callus per treatment".
Comments 26: What references was the optimization based on?
Response 26: We have added related references to this section.
Discussion:
Comments 27: As an introduction, the discussion is too long and too general and discuss cryopreservation at some parts general - and this is boring.
It should focus on (1) vitrification-dehydration and on (2) cryopreservation of Anthurium.
Response 27: We have modified the discussion especially on vitrification and cryopreservation of Anthurium, and also added the reference mentioned below.
There is another reference on cryopreservation not mentioned in the article:
Hanan, M. A., & Sherif, S. S. (2013). Conservation Plantlets of Anthurium andraeanum (Lindeu ex Andre) By using in Vitro. Middle East j, 2(2), 36-43.
Comments 28: There is no conclusion!
Response 28: We have added the conclusion.
Reviewer 3 Report
Comments and Suggestions for Authors
This manuscript describes the cryopreservation procedure for Anthurium andraeanum callus, using multi-component testing experiments in the preculture, loading and dehydration phases. The authors modified the preculture temperature and the composition of the loading and vitrification solutions. The callus was rapidly frozen in liquid nitrogen and then rapidly thawed. After one day of recovery, the efficacy of the surviving cells was determined using the TTC procedure. The best cell survival was obtained using the following protocol (in brief): preculture at 25 °C for 2 days, loading in 50% of vitrification solution (with glycerol, ethylene glycol and anothers component) at 25 °C for 60 min, dehydration at 0 °C for 60 min with a solution containing 0.08 mM reduced glutathione, rapid cooling in liquid nitrogen 72 for 1 h, followed by rapid warming at 40 °C for 90 s and unloading for 30 min.
The presented manuscript should be considered as a the methodological paper. However, the topic of the presented research is interesting and valuable for both basic research and application. The presentation and discussion of the results are correct. In my opinion, the manuscript needs a minor revision after the improvement according to the remarks and questions mentioned below:
1) Do the authors know what type of callus was obtained on the medium described in the Materials and Methods? Is it callus with morphogenic potential? What type of protocol (previously published or not) was used to induce callus?
2) Line 215-216, the sentence is incomplete: Over the past two decades, vitrification-based cryopreservation of A. andraeanum using suspension cells and axillary buds as materials, respectively, has been developed.
3) Lines 433 and 437, could the Author explain what the “CK treatment” is?
4) Line 451, The full name of the plant growth regulator abbreviated TDZ is thidiazuron, not thifensulfuron!
Best regards.
Author Response
Comments 1: Do the authors know what type of callus was obtained on the medium described in the Materials and Methods? Is it callus with morphogenic potential? What type of protocol (previously published or not) was used to induce callus?
Response 1: We agree with this comment. The experimental material is non-embryonic callus but can be induced adventitious bud. It can be seedling-forming if it continues to be induced. We have added the methods of inducing callus, which are not published yet.
Comments 2: Line 215-216, the sentence is incomplete: Over the past two decades, vitrification-based cryopreservation of A. andraeanum using suspension cells and axillary buds as materials, respectively, has been developed.
Response 2: Thank you for pointing this out. We have modified this sentence.
Comments 3: Lines 433 and 437, could the Author explain what the “CK treatment” is?
Response 3: Thank you for pointing this out. CK is the procedure without adding exogenous substances (treatment 10 with LS2), and we have explained it in the manuscript.
Comments 4: Line 451, The full name of the plant growth regulator abbreviated TDZ is thidiazuron, not thifensulfuron!
Response 4: Thank you for pointing this out. We have modified this word.